# The regulatory effect of herd structure on pig production under the environmental regulation

**Gangyi Wang, Yuzhuo Shen\*, Chunlei Li, Qiuping Zhu, Aidyn ZhanBota**

School of Economics and Management, Northeast Agricultural University, Harbin, China

\* shenyuzhuo@neau.edu.cn

**Data Availability Statement:** Third-party data is used in our article. We will provide the sources and query methods of all Third-party data and all the data we used for analysis for repeated verification. All data we use are derived from the following

## Abstract

Integrating pig production stability and pollution control is a matter of social livelihood and green development. This paper estimates the policy effectiveness of environmental regulation based on the big data of Chinese government websites and combines the pig production data of various provinces in China from 2008 to 2018 to construct a mediating effect model, empirically analyzing the relationship between environmental regulations, herd structure, and pig production. Research shows: (1) Improving environmental regulations in the short term will increase the slaughter of pigs, but in the long run it will decrease first and then increase; (2) At this stage, environmental regulations can increase the slaughter of pigs by changing the herd structure; (3) Reasonable adjustment of the herd structure can effectively increase the slaughter of pigs; (4) The herd structure can be used as a supplementary monitoring indicator to stabilize the supply of pigs. Combine it with the change rate of sow stock to provide early warning of fluctuations in pig production, and the early warning herd structure values are 0.0980 and 0.1135. There are two key initiatives to achieve supply and price stability in the pig industry that taking into account environmental protection: first, an industrial regulation instrument based on the herd structure needs to be established; second, an early warning system for pig production fluctuations should be established with reference to the early warning system for pork price fluctuations.

## 1. Introduction

Stabilizing pig production and ensuring the supply of pork is related to the development of "agriculture, rural areas and farmers", people's lives, and the overall situation of economic and social development. Since 2019, pig prices are affected by many factors such as macroeconomic policies, epidemics, regional differences, etc. However, the imbalance of supply and demand is the root cause of price fluctuations in the pig market [1, 2]. After the promulgation of the "Regulation on the Prevention and Control of Pollution from Large-scale Production of Livestock and Poultry" (restriction order) in 2013, the stock of pigs and sows has gradually decreased at an average annual rate of 2.34% and 3.63%. Although the outbreak of ASF in China in 2019 had a severe impact on pig and sows production, in December 2019, the

sources: 1. The official websites of the Chinese government (http://www.gov.cn/) and the provincial governments,such as(https://www.sc.gov.cn/; http://www.zj.gov.cn/; http://www.gd.gov.cn/; http://www.fujian.gov.cn/; http://www.shandong.gov.cn/; http://www.ln.gov.cn/; http://yn.gov.cn/; http://shanxi.gov.cn/; http://cq.gov.cn/; http://jl.gov.cn/; http://sdgb.shandong.gov.cn/; http://www.hebei.gov.cn/; https://www.ah.gov.cn/; http://www.jiangxi.gov.cn/; http://www.shaanxi.gov.cn/; http://www.gansu.gov.cn/; https://www.hubei.gov.cn/; http://www.jiangsu.gov.cn/; https://www.nmg.gov.cn/; https://www.hainan.gov.cn/; http://www.qinghai.gov.cn/; http://www.jiangxi.gov.cn/; https://www.hlj.gov.cn/index.shtml; http://www.ln.gov.cn/; https://www.henan.gov.cn/; https://www.xinjiang.gov.cn/; https://www.guizhou.gov.cn/; http://www.cq.gov.cn/); 2. Statistical Yearbook published in China. Specifically including the National Agricultural Products Cost and Benefit Information Collection (ISBN: 978-7-5037-9038-6), CHINA ANIMAL HUSBANDRY AND VETERINARY YEARBOOK (ISSN: 2095-9966), and CHINA STATISTICAL YEARBOOK (ISBN: 978-7-5037-9225-0) 3. Unpublished materials shared by the network, Handbook on Production and Discharge Coefficient and Discharge Coefficient of Livestock and Poultry Breeding Industry in the First National Pollution Source Census (https://max.book118.com/html/2021/0617/8076070056003111.shtm).

**Funding:** This study was funded by the Humanities and Social Science Foundation of Ministry of education of China, Research on Protection and Utilization of Pig Genetic Resources in China: Organization, Efficiency and Innovation [21YJA790053], the Natural Science Foundation of Heilongjiang (CN), Research on Capital Stress and Counseling of Pig Industry in Heilongjiang Province under Environmental Regulation [LH2019G002], the Chongqing Yingcai Program Social Science Planning Project Research on Identification of Safety Boundary, Driving Factors and Upgrading Path of Pig Industry, and the Northeast Agricultural University Talent Program in the form of grants and funds to GW.

**Competing interests:** The authors have declared that no competing interests exist.

National Conference of Directors of Agricultural and Rural Departments proposed that "we cannot go back to the old path of sloppy development and sacrificing the environment because of a shortage", emphasizing that the recovery of the pig market cannot be at the expense of the environment. It can be seen that ensuring the cleanliness of the produce process and controlling the pollution of breeding are not only the core means to achieve sustainable development, but also an important prerequisite for the development of the pig industry. Continuous and targeted environmental regulations are required to promote the green development of the pig industry. Therefore, clarifying the impact of environmental regulations on pig production is of strategic and developmental importance for securing the market supply of meat products, improving the environment of rural residents, and green upgrading of the livestock and poultry industry.

Existing literatures mostly analyze the impact of environmental regulation on pig industry from two aspects: environmental regulation on production and industrial risk response ability [3]. Neo-classical theory holds that the cost of aquaculture enterprises will increase and the output will decrease under environmental regulations, which will lead to oversupply of the market [4]. Azzam A et al. used a comparative static model to analyze the impact of strict environmental regulations on the production of different types of farms in the United States, indicating that the production of large farms under environmental regulations will decrease [5]. Further research by Zhang LY et al. shows that the impact of environmental regulation on pig production will also affect people's livelihood through the cascade amplification of industrial chain and Consumer-Price-Index (CPI) [6].

Behavioral economic theory holds that market supply is determined by enterprise output, and the sum of production of farms under environmental regulation is market supply. Yang Haotian et al's MOA model and two-column estimation analysis show that strengthening environmental regulation will improve farmers' awareness of environmental pollution and encourage farmers to carry out cleaner production [7]. Xu Lifeng et al. used Logit and Tobit regression models to further analyze the differences in production behavior of different scale farms under environmental regulations, and believed that scale enterprises were more inclined to cleaner production [8].

The above studies quantify environmental regulation as intensity index, and then analyze the impact of regulation on pig production. However, the actual carrier of environmental regulation is the policy text, and its function has a certain lag. Chen S et al. used the Difference-in-Difference (DID) model to analyze the cost and benefit of environmental regulation policy for China's pig industry, which shows that the regulations significantly reduced $NH_3$-N, but also the pig industry shrank significantly, resulting in a loss of 2.9% of China's agricultural output value [9].

The impact of environmental regulations on pig production is also affected by price buffer space and changes in policy uncertainty. Bowen C et al. used the data of 76 farmers' markets in Africa, Asia and Latin America from 2000 to 2015 to analyze the pig production when the domestic market environment changes, showing that international trade can only solve the problem of insufficient production within a certain range [10]. Li PC et al. analyzed the impact of China's environmental policy and African swine fever on production, showing that when the policy is uncertain, the enterprises will face greater risks, and will reduce production or even withdraw from the market, resulting in insufficient supply in the market [11]. Existing studies have fully discussed the inhibitory effect of environmental regulations on pig production, but is it true that environmental regulations are not compatible with production as a basic national policy? Whether the adjustment of key variables can achieve the balance of clean production and supply and price stability has become an issue that needs to be studied urgently.

After reviewing the existing studies, it is found that although the studies on the analysis of the impact of environmental regulations on pig production are relatively abundant, there are still two deficiencies in the following two aspects, which urgently need to be strengthened and supplemented. First, most of the existing studies based on static analysis believe that environmental regulations have an inhibitory effect on pig production, and the impact of environmental regulations on pig production needs to be further clarified from a dynamic perspective. Second, the existing research and analysis of the impact of environmental regulations on the industry are mostly based on the behavior of the main body and the regulatory means, and lack of control means for the industry itself. It is necessary to start from the pig industry to explore feasible solutions that can take into account environmental protection and pig production. Based on this, while analyzing the impact of environmental regulations on pig production, this paper introduces the lagging variables of environmental regulations into the model and analyzes the impact of environmental regulations on pig production from a dynamic perspective. Meanwhile, this paper incorporates the herd structure into the analysis framework and analyzes the role of the herd structure in the impact of environmental regulations on pig production. The theoretical and empirical contributions of this paper are as follows:

First, our research comprehensively considered the hysteresis effect of environmental regulations and the U-shaped relationship, and change the impact of environmental regulations on pig production from a static effect to a dynamic effect.

Second, our research took the herd structure as an entry point, incorporate it into the analysis framework of the impact of environmental regulations on pig production, and analyze the impact of the herd structure on pig production under environmental regulations.

Third, our research compared the impact of the pig grain ratio and price, put forward the possibility of the herd structure as an early warning indicator of pig supply, and set an early warning interval based on its fluctuations in production.

## 2. Theory and hypotheses development

### 2.1. Pig supply model with herd structure under environmental regulations

Factors of production refer to all the elements and environmental conditions necessary for material production. When analyzing product supply, most studies are based on the C-D production function, constructing a universal supply model, and analyzing the impact of capital, labor, and other factor inputs on production [12]. In the animal husbandry industry, some studies separately analyze factors such as land, epidemic prevention, feed, dams, and young animals to examine the impact of specific factors on production [13, 14]. In order to analyze the impact of herd structure on pig supply, this paper takes sows as a production factor, constructs a pig supply model, and analyzes the relationship between herd structure and pig production under environmental regulations. In the self-breeding and self-raising model, piglets are not only a production factor in the fattening process but also a product that can reproduce sows during the produce process. When reproducing sows and piglets are included in the same framework, it is necessary to analyze the feasibility of both sows and piglets as factors of production at the same time.

First, as an important production factor in the produce process, sows have higher purchase costs and maintenance costs than piglets. The input as a production factor is closer to fixed asset investment. There are mature sows insurances and loan products that use sows as collateral, which shows that sows have better stability in the production process. Second, the production process aimed at traditional products is also longer than piglets. In short-term production, the sows's products are only piglets. Only when the gestation cycle of the sows and the fattening cycle of the piglet is considered at the same time, its products are traditional

products such as slaughter pigs [15, 16]. Third, in actual production, in addition to self-repro-duction and self-support, there are also a large number of companies that specialize in fatten-ing or piglet produce. For companies specializing in piglet produce, sows as a production factor are only piglets, and pigs for slaughter are not used as the final product. Based on the above three aspects, the reproductive sows and piglets are used as production factors at the same time to analyze the impact of the adjustment of the herd structure on production in the short term.

In the field research, facing external shocks such as environmental regulation, the major producers will give priority to adjusting the stock of fattening pigs in order to quickly meet the environmental protection requirements, and obtain certain cost compensation through early sales to reduce losses. However, with the increase of external shocks, they will also kill sows. Based on the above facts, when analyzing the adjustment of production input factors, it is assumed that the main produce subject will give priority to adjusting the input of piglets and set short-term and long-term target differences based on whether sows are adjusted or not. The main produce subject generally does not adjust the farming output of sows in the short term, but under the guidance of long-term goals, the main produce subject will strategically adjust sows as a production factor that must be considered separately (Fig 1).

Chinese pig production is mainly self-reproduction and self-raising. After considering the contractual relationship between the upstream and downstream of the industrial chain, the non-self-reproducing and self-supporting enterprises can still be approximated as self-repro-ducing and self-supporting entities for analysis. This kind of subject needs to input the sows $X_1$ and other production factors $X_2$ including piglets and feed input in the piglet fattening link to produce with the production function $Y = f(X_1, X_2)$. Its production cost is $C = w_1X_1 + w_2X_1$, Where $w_i$ is the used price of production factor $X_i$, the rational peasant theory believes that the producers will take profit maximization and cost minimization as the standard production. Refer to the government's mandatory environmental regulation conditions set by You J et al. [17] to analyze the production situation of the producers under different environmental regu-lations, and mark different situations with t. Set $t = 0$ to indicate that the environmental regula-tion has not been carried out, $t = 1$ is the situation of environmental regulation in the short term, $t = 2$ is the situation of environmental regulation in the long term, and $t = 2+k$ is the con-text in which the intensity of environmental regulation changes in the long term. The main

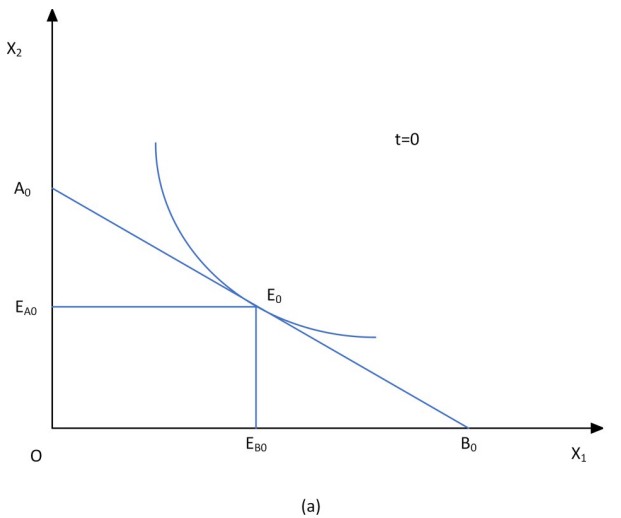 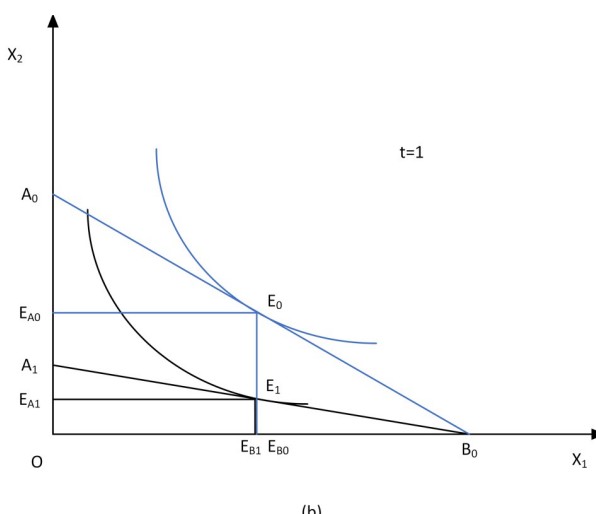

(a)                                                    (b)

**Fig 1. The changes in the inputs and outputs of the main produce subject.**

produce subject produces at the intersection $E_0$ of the equal-cost line and output isoquant, and the optimal factor input at this time is $E_{A0}$ and $E_{B0}$ respectively, at $t = 1$, environmental regulation only affects the production factors $X_2$, and at $t = 2$, environmental regulation can affect both production factors $X_1$ and $X_2$.

Environmental regulation needs to be concrete as a specific policy tool before it can be applied to pig production. Existing environmental regulation tools include technical level restrictions, total pollution restrictions, and environmental tax on a specific duty, among which technical level restrictions are mainly enforced by national administrative coercive and are hard standards, while total pollution restrictions and environmental tax on specific duty need to rely on market mechanisms, concretely manifested in emission rights and environmental taxes, which are typical marketization environmental regulation tools [18]. Since this paper focuses on the impact of environmental regulation on pig production under the market mechanism, only the two situations of environmental tax collection and the issuance of pollution rights will be explained.

## 2.2. Non-adjustable factors of production began to adjust

Both the imposition of environmental taxes and the issuance of emission rights require additional costs for factors of production for the main produce subject, which is equivalent to an increase in factor prices. When the production factor cannot be adjusted freely *(t = 1)*, only the production factor $X_2$ changes and the equal-cost line changes from $A_0B_0$ to $A_1B_0$, where $A_0A_1$ is the additional cost of the main produce subject under environmental regulation, in actual production, this is mainly expressed as the cost of environmental taxes or the purchase of emission rights (Fig 1). The enterprise still produces in the way of maximizing profit and minimizing cost, and its output level is still the tangent point of the isoquant line and the equal-cost line, that is point $E_1$. At this time, there must be a downward shift of the isoquant line of pigs, that is, the output decreases. At this time, the optimal factor input is $E_{A1}$ and $E_{B0}$ respectively. In actual production, the producers can reduce production by producing fattening pigs for slaughter and reducing piglets. Among them, slaughter pigs will increase the supply of pigs in the market in the short term and decrease in the long term; reducing piglets will not have an impact on the supply of pigs in the short term, but it will reduce the long-term supply of pigs.

Hypothesis 1: when other conditions remain unchanged, strengthening environmental regulation will lead to a short-term increase and a long-term decrease in the slaughter of pigs.

When the factors of production can freely adjust the *(t = 2)*, considering the substitution effect between production factors, the organic composition of capital can be adjusted (Fig 2). The production factor $X_2$ is simplified as the input of piglets in the production process to analyze the impact of herd structure and the production of pigs. The main produce subject needs to adjust the input of factors to achieve maximum profit and minimize cost. Since the prices of both the factor $X_1$ and the factor $X_2$ have changed, they are recorded as $w_i + \theta_i$, where $\theta_i$ is the environmental tax paid by the farming subject for the production factor $i$. According to the "Taxable Pollutants and Equivalent Value Table", sows and fattening pigs pay the same environmental tax during the fattening process, namely $\theta_1 = \theta_2$. At this time, the factor input ratio of the optimal output should be $(w_1 + \theta_1)/(w_2 + \theta_2)$, which has a greater slope of the equal-cost line than when $t = 1$ close to -1, as shown in $A_1'B_1'$ in Fig 2c1. At this time, $A_1A_1'$ and $B_0B_1'$ are the adjustment of production factors $X_2$ and $X_1$, respectively according to the weak axiom of cost minimization (WACM), the change of factors from optimal to non-optimal under equal output will inevitably lead to increased costs. With constant cost constraints, changes in the factors of production toward the optimal combination will promote an increase in output, that

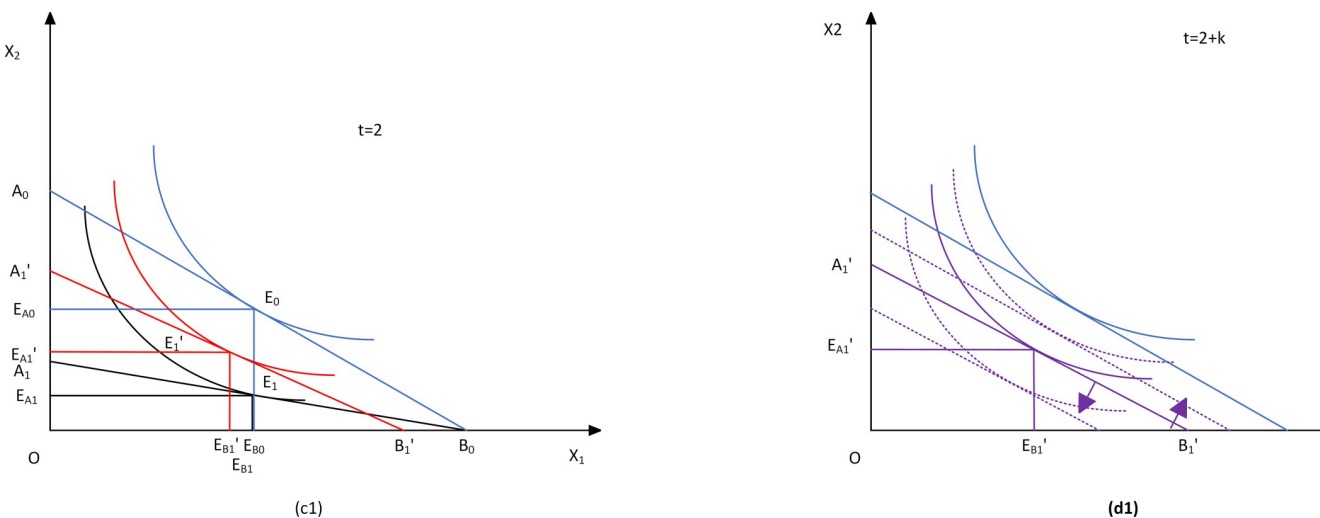

**Fig 2. The effect of whether the factors of production are adjustable on the subject's input and output under environmental regulation.**

is, under the condition that the main produce subject does not increase the input, freely adjusting the configuration of the production input elements $X_1$ and $X_2$ can increase the output.

The regulation of total pollutant is different from the collection of environmental tax. The government does not price the pollution generated during the use of every factor of production. Therefore, the main produce subject can produce according to the initial factor price ratio. At this time, the condition for maximizing profit is $w_1/w_2$. As shown by $A_1'B_1'$ in Fig 2d1, $A_1A_1'$ and $B_0B_1'$ are the adjustment of production factors $X_2$ and $X_1$, respectively. From the cost minimization axiom, it can be seen that during the produce process, the input elements $X_1$ and $X_2$ are allowed to be adjusted, which can increase the output without increasing the main input. Since the total pollution regulation does not change the optimal production conditions, the equal cost line is translated rather than rotated when the total pollution regulation is implemented, that is, pig production only depends on the initial price of the factor.

Hypothesis 2: Under the established production cost of the main produce subject, reasonable adjustment of herd structure can improve pig production under the intensity of specific environmental regulation.

## 2.3. Pig production changes under environmental regulations

Considering the impact of environmental regulation intensity change on pig production and set it to $t = 2+k$ periods. When the government levies an environmental tax, the produce subject adjusts the inputs of production factors according to the ratio of environmental tax levied on production factors to make them produce under optimal conditions, that is, to ensure that the ratio of factor inputs is equal to $(w_1 + \theta_1)/(w_2 + \theta_2)$, when $\theta_i$ increases as the intensity of environmental regulation increases, the cost for pig production decreases if the total cost of farming remains the same so that the equal-cost line is shifting flat to the lower left. Under the current levy standard $\lim\limits_{\theta_i \to \infty} \frac{\theta_1}{\theta_2} = 1$, which means that the equal-cost line gradually approaches -1 in the process of translation, the expansion line of the main produce subject with the change of environmental regulation intensity is shown as OQ (Fig 3).

When the government regulates by selling the total amount of emission rights, it is assumed that the market sells the emission rights in an unlimited amount, and the production function of the main produce subject is shown in Fig 1A. When the intensity of environmental

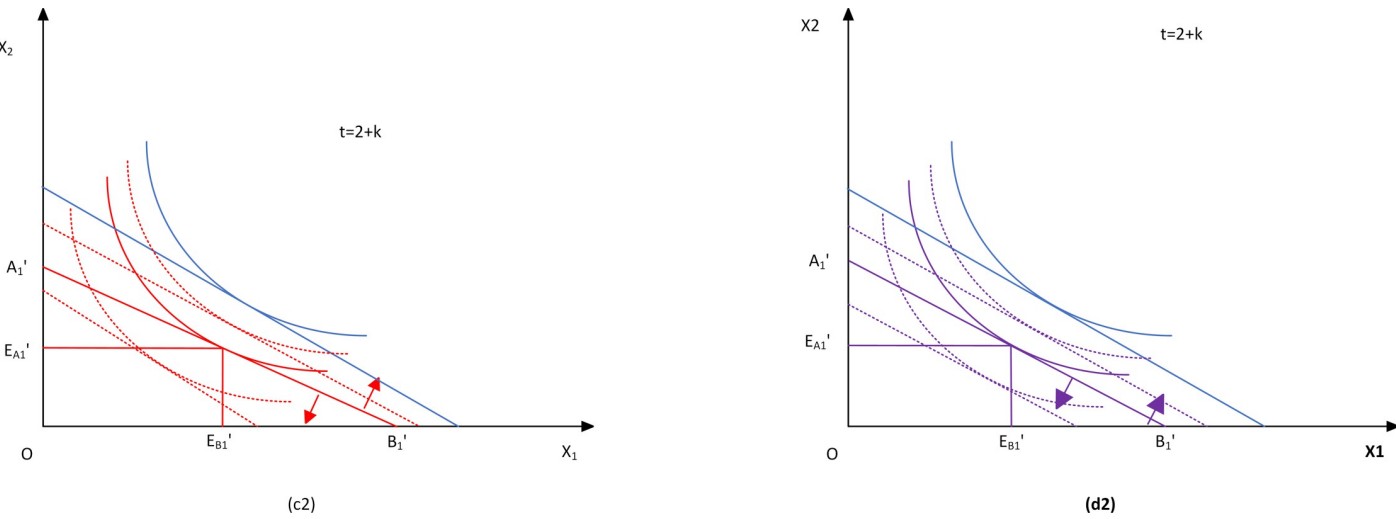

**Fig 3. The effects of changes in the intensity of environmental regulation on the input and output of the subject.**

regulation increases and the emission rights $\tau \rightarrow 0^+$ are sold, the main produce subject will still produce as described in d1 when it produces, that is, they will approach the origin in a translational manner. At this time, the expansion line of the main produce subject with the intensity of environmental regulation is shown as *OQ* in Fig 3*d2*. The intensity of environmental regulation and the herd structure respectively affect the production of pigs, and the intensity of environmental regulation will also affect the production of pigs through the herd structure.

Hypothesis 3: the herd structure has a mediating effect on the impact of environmental regulations on pig production.

## 3. Methodology and data

### 3.1. Models and methodology

**3.1.1. Methods for estimating the effectiveness of environmental regulation policies.**
Policy effectiveness can comprehensively reflect the strictness of environmental regulatory policies, which is one of the important characteristics of environmental regulatory policies. However, since the effectiveness of the policy is difficult to measure, it is necessary to subdivide the indicators [19]. When analyzing China's energy conservation and emission reduction policies, Zhang G [20] puts forward indicators for the effectiveness of environmental regulatory policies, and it is also decomposed into three dimensions of policy strength, policy objectives, and policy measures for scoring. With the rise of environmental decentralization theory and environmental federation theory, local governments have more discretionary powers in environmental regulation. However, it still needs to be monitored by the central government due to the promotion mechanism [21]. Therefore, this paper introduces policy feedback from indicators measuring environmental supervision into the policy effectiveness evaluation model. Refer to Pan D et al. [22] 's evaluation criteria for environmental feedback, the four aspects of policy strength, policy objectives, policy measures, and policy feedback are evaluated as shown in Table 1.

In Table 1, policy objectives, policy measures, and policy feedback are used to reflect the effectiveness indicators at the policy formulation level, and the policy strength reflects the effectiveness indicators at the policy implementation level, which are used to ensure that the policy achieves the expected goals. In this paper, based on the relationship among policy

objectives, measures, feedback, and strength, the policy effectiveness evaluation model is set as follows.

$$Hjgz_{iP} = E_{acc_{iC}} + E_{acc_{iP}} \tag{1}$$

$$E_{acc_i} = \sum_{j=1}^{m} S_j \left( O_j + M_j + F_j \right) \tag{2}$$

Among them, $Hjgz_{iP}$ is the effectiveness of environmental regulation in the year $i$ of Province $P$, which is used to reflect the stringency of environmental regulation. $E_{acc_{iC}}$ is the cumulative effects of environmental regulatory policies at the national level, and $E_{acc_{iP}}$ is the cumulative effects of environmental regulatory policies at the provincial level in China, $S$ means policy intensity, $O$ is policy objective, $M$ is policy measure, $F$ is policy feedback, and $j$ is the specific environmental regulation policy introduced in the year $i$.

**3.1.2. Econometric model setting.** The existing studies mostly use the step method to analyze the mediating effect [23], This paper builds the mediating effect model based on this,

**Table 1. Criteria for evaluating the effectiveness of China's environmental regulation policies.**

| Indicator Categories | Scores | Standards |
|---|---|---|
| | 5 | Specific and clear objectives, quantifiable, with clear quantitative criteria |
| policy objective | 4 | Specific goals, vague quantitative criteria |
| | 3 | Specific objectives, not quantified (e.g. reforestation, haze control, etc.) |
| | 2 | The class targets (e.g. air pollution control, water pollution control, control of domestic waste, etc.) |
| | 1 | Visions (e.g. pollution control, ecological protection, etc.) |
| | 5 | List specific measures, give strict implementation and control standards and specify |
| policy measure | 4 | List specific measures, give more detailed implementation and control standards |
| | 3 | List the more specific measures, give the general implementation content from many aspects of classification |
| | 2 | List some basic measures and give a brief implementation content |
| | 1 | Only talk about the relevant content from a macro perspective, no specific operation plan |
| | 5 | Specific department in charge, regular feedback |
| policy feedback | 4 | Specific responsible department, non-regular feedback |
| | 3 | Specific department in charge, no feedback |
| | 2 | No specific department in charge |
| | 1 | No supervision |
| | 5 | Laws |
| policy intensity | 4 | Decisions, Opinions, Notices, Regulations, and Provisions of the State Council of the Central Committee of the Communist Party of China |
| | 3 | Provisional Regulations and Provisions promulgated by the State Council, Regulations and Provisions promulgated by each part |
| | 2 | Opinions, measures, implementation plans promulgated by ministries and departments of the State Council |
| | 1 | Circulars, plans, local government policies, and non-governmental organization policies issued by various ministries and departments of the State Council |

as shown in formula (3).

$$
\begin{aligned}
Y &= cX + e_1 \\
M &= aX + e_2 \\
Y &= c'X + bM + e_3
\end{aligned}
\tag{3}
$$

Among them, $Y$ is the explained variable, $X$ is the core independent variable, $M$ is the mediator variable, $a$, $b$, and $c$ are the regression coefficients, and $e$ is the residual.

Since the number of pigs for column quantity is usually affected by the expectations of the producers when analyzing the production of pigs, it is necessary to consider the impact of pig production itself, that is, a dynamic panel model should be constructed. To ensure the consistency of the estimation, the system Gaussian Mixed Model (GMM) is selected for estimation, and the econometric model for analyzing the impact of environmental regulation on pig production is as follows.

$$
lnSzcl_t = \alpha_0 lnSzcl_{t-1} + \alpha_1 lnHjgz_t + \alpha_2 lnYzjg_t + \alpha_4 lnWzf_t + \alpha_5 lnRgcb_t + \alpha_6 lnTdcb_t + \mu \tag{4}
$$

The mediating effect model is as follows.

$$
\begin{aligned}
lnSzcl_t &= \alpha_0 lnSzcl_{t-1} + \alpha_1 lnHjgz_t + \alpha_2 lnYzjg_t + \mu \\
lnYzjg_t &= \beta_0 lnYzjg_{t-1} + \beta_1 lnHjgz_t + \mu
\end{aligned}
\tag{5}
$$

Among them, $Szcl$ is the pig production, $Hjgz$ is the environmental regulation, $Yzjg$ is the herd structure, $Wzf$, $Rgcb$ and $Tdcb$ are control variables.

## 3.2. Variables selection and measurement

**3.2.1. Dependent variable: Pig production.**   This article uses the column quantity to measure the production level of pigs, as the explained variable of this article. pig production usually includes many aspects such as produce scale, herd structure, technical level, etc. However, the column quantity is the basis for measuring the level of pig production. Therefore, this article uses the column quantity data of the China Animal Husbandry Statistical Yearbook to measure the level of pig production in various provinces.

**3.2.2. Independent variable: Pig production.**   Environmental regulation. This paper uses the effectiveness of environmental regulation policies to measure the intensity of environmental regulation as the core explanatory variable of this paper. After 2008, the Chinese Government Network began to actively publicize the cur-rent policies. Although there are some early policy announcements, the number of announcements is significantly smaller than after 2008. Taking into account the impact of policy deficiencies on environmental regulation, this paper selects relevant data from 2008 to 2019 for analysis.

According to the above formulae, this paper crawls big data for the website of the Central People's Government of the People's Republic of China and 31 official government websites of Chinese provincial units. In the site selection, the crawl uses the keyword search method to search, set the keywords as "pollution", "environment", and "ecology", crawl the policies published from January 1, 2008, to December 31, 2019, and obtain data in total 247,315, and 13429 related policies for the environmental regulation of the pig industry were obtained after screening, and the screening steps are shown in Table 2.

Through the previous policy effectiveness evaluation, the results are shown in Fig 4.

In general, the effectiveness of environmental regulation policies for the pig industry began to increase substantially after 2013. Although it declined in 2018, it has reached a high level. It is generally believed that environmental regulation for the pig industry has been gradually

**Table 2. Screening steps of environmental regulation policies.**

| Steps | concrete contents |
|---|---|
| 1 | Eliminate duplicate data. |
| 2 | Retain the terms "breeding", "animal husbandry", "pig", "column", "feed", "fattening", etc. "feed", "fattening" and other words related to the pig industry appear in the text. |
| 3 | Keep the words "ecological environment", "ecological civilization", "environmental protection", "resource environment", "pollution", "resource bearing", "natural resources", "protection of the environment", "environmental management", "ecological protection", "recycling", "environmental ecology", "natural environment", "ecological construction", "ecological priority", "green development", " human habitat", "ecological benefits", "ecological restoration", "environmental remediation", "environmentally friendly", "environmental cleanliness", "environmental standards" and other words appearing in the text. |
| 4 | Exclude the words "development environment", "competitive environment", "teaching environment", "social environment", "academic environment", "investment environment", "trading environment", "consumer environment", "international environment", "market environment", "financing environment" and "ion environment" from the text. |
| 5 | The remaining entries are identified manually, and the identification principles are as follows: (i) they are documented entries such as laws, regulations, and policies issued by the government with clear document numbers; (ii) the documents are directly or indirectly related to environmental protection and pollution control; (iii) the purpose of the policy documents is pollution control and environmental protection, rather than a description of the situation based on the existing environment. |

strengthened since the State Council issued the Livestock and Poultry Scale Pollution Prevention Regulation in 2013. With the revision of the "Environmental Protection Law" in 2015 and the introduction of the "Guiding Opinions on Promoting the Adjustment and Optimization of the Distribution of Pig Production in Southern Water Net Region " in 2016, the environmental regulation of the pig industry has reached a new height [24, 25]. The effect of the pig industry environmental regulation policy estimated in this paper is the same as this trend, and it can be used as the main indicator to measure the intensity of environmental regulation. Comparing the composition of the effectiveness of environmental regulation policies, the policy feedback and policy objectives are significantly higher than the policy measures and policy intensity.

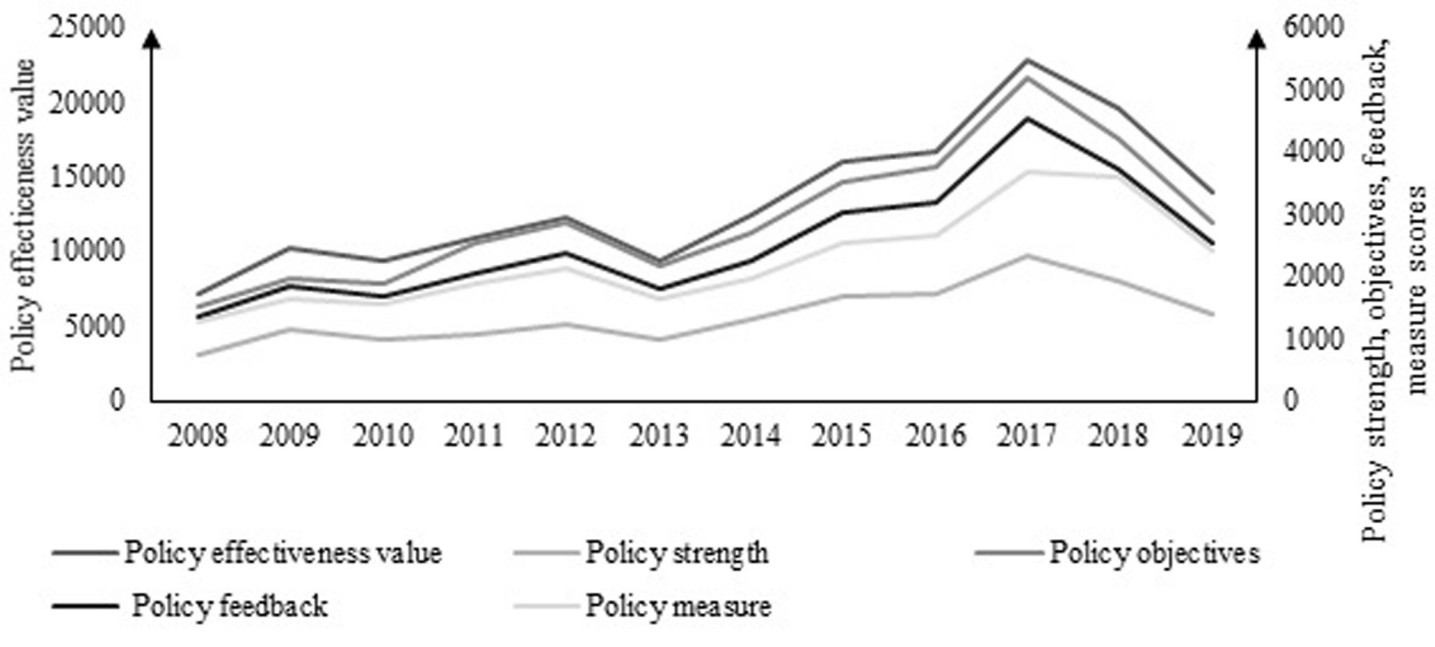

**Fig 4. Trends in the effectiveness of China's pig industry environmental regulation policies.**

The main reason for this may be because the new environmental protection law specifically strengthens the responsibility for environmental protection at all levels of government. To meet this standard, environmental regulatory policies need not only to refine regulatory goals but also to clarify the responsible units for each objective goal, so that the promotion of policy goals and policy feedback is significantly higher than other components of the effectiveness of environmental regulation policy effectiveness.

**3.2.3. Mediator variable: Herd structure.** The herd structure includes three main parts: pigs, sows, and finishing pigs. Gao Y et al. [26] take the sows as the core, construct the pig group number to measure the herd structure of the farm, set the number of pig groups as the ratio of the produce time of the pigs in the corresponding barn to the reproduction rhythm, where the reproductive rhythm involves multiple indicators such as the number of farrowed sows, the number of pregnant sows, the number of sows, the number of suckling piglets, the number of weaning piglets, and the number of fattening pigs. Although this method can effectively identify the herd structure, it is more suitable for the internal production management of the main produce subject due to the complexity of calculation and the short time interval of data requirements. Based on the core definition of the herd structure, this article uses the proportion of the sows to the total sows and fattening pigs as a specific indicator to describe the herd structure for industrial level analysis. The calculation method is as follows.

$$Yzjg = \frac{Sows}{Sows + Pigs} \tag{6}$$

Among them, Sows is the number of stock sows at the end of the year, and Pigs is the number of stock pigs at the end of the year. The above data are from *the China Animal Husbandry Statistical Yearbook*.

**3.2.4. Control variables.** In neoclassical economics, the production of products is related to the demand of the product market and the price of the factor market. Therefore, this paper sets control variables from the above two perspectives. Since the market demand for pigs cannot be obtained directly, this paper selects and uses the market pig price as a proxy variable to measure demand for analysis and the factors of pig production are divided into three parts: land, labor, and capital. Select the 50 kg pig column price, land cost, labor cost, and material service fee in *the China National Compilation of Cost and Income of Agricultural Products* as specific indicators.

The theory of external shocks holds that policy changes are external shocks that affect production. However, for pig production in China, the epidemic situation is also an important external impact. Especially, the outbreak of African swine fever in China in 2019 led to a sharp drop in China's production capacity. In order to eliminate the influence of this factor on the regression results, we set a virtual control variable ASF to indicate whether there is an African swine fever epidemic. When the variable ASF is 0, there is no African swine fever epidemic, and when the variable ASF is 1, there is an African swine fever epidemic.

In addition, because *the China National Compilation of Cost and Income of Agricultural Products* does not include statistics on the cost and benefit of Tibetan pigs, and the data for Beijing and Tianjin in 2019 are missing. Therefore, in addition to Beijing, Tianjin, and Tibet, data from 28 provinces in China from 2008 to 2019 were selected for analysis. We use the 2008 CPI as the base period to reduce the price-related data. The statistical description results of each variable after reduction are shown in Table 3.

**Table 3. Selection of indicators for the variables of the empirical analysis.**

| Category of variables | variables | | symbol | unit | Obser-vations | mean | scl | Min-imum | Max-imum |
|---|---|---|---|---|---|---|---|---|---|
| Dependent variable | Pig production | | *lnSzcl* | Million heads | 336 | 7.34 | 1.1063 | 4.52 | 8.92 |
| Independent variable | Environmental Regulation | | *lnHjgz* | - | 336 | 7.37 | 0.5349 | 5.99 | 8.25 |
| Mediator variable | Herd structure | | *lnYzjg* | - | 336 | -2.35 | 0.1350 | -2.63 | -1.87 |
| Control variables | Pig price | | *lnSzjg* | Yuan | 336 | 6.44 | 0.4853 | 4.28 | 7.38 |
| | Factor input | Capital | *lnWzf* | Yuan | 336 | 7.05 | 0.5423 | 4.92 | 12.69 |
| | | Labor | *lnRgcb* | Yuan | 336 | 5.17 | 0.9836 | 2.22 | 6.67 |
| | | land | *lnTdcb* | Yuan | 325 | -0.37 | 1.3529 | -5.66 | 2.53 |
| | Disease impact | | *ASF* | - | 336 | 0.08 | 0.0766 | 0 | 1 |

# 4. Empirical results and analysis

## 4.1. Pre-regression test

Panel data analysis requires data to be stable to ensure that there is no spurious regression in the regression results. Commonly used detection methods include Harris-Tzacalis(HT) test, Levin-Lin-Chu(LLC) test, and Im-Peraran-Shin(IPS) test, respectively for the short panel, long panel, and less samples [27]. Therefore, this article is subject to HT test and IPS test. The specific test results are shown in Table 4.

According to the results of the HT test, the *lnSzcl*, $lnHjgz_t$, $lnYzjg_t$ and *lnRgcb* did not pass the unit root test, that is, the data is non-stationary, and there may be spurious regression in the direct regression analysis. The existing methods include differential processing and cointegration test. Because the differential processing will change the original economic meaning of data, this paper gives priority to the cointegration test Commonly used cointegration test methods include the KAO test, Pedroni test, and WesterLund test. The specific results are shown in Table 5.

According to the results of the cointegration test, the three test results all show that the variables of the econometric model have a cointegration relationship, cointe-grated data can be directly regressed without spurious regression [28].

## 4.2. The impact of environmental regulations on pig production

System GMM estimation requires the Sargan test (over-identification) and the Abonda test (sequence correlation), the two-stage system GMM estimation is selected according to the test results, as shown in Regression 2. In the Regression 2, the coefficient of the $lnSzjg_t$ is -0.0567, that is, the slaughter of pigs in the t period will increase with the decrease in the price of slaughter, which is inconsistent with the actual production, which may be caused by the inversion of cause and effect. Zhou J et al. [29] come to a similar view when analyzing the fluctuations in

**Table 4. Unit root test results.**

| | HT test | LLC test | IPS test |
|---|---|---|---|
| $lnSzcl_t$ | 0.9843 | 0.0020 | 0.1816 |
| $lnHjgz_t$ | 0.3617 | 0.0770 | 0.9925 |
| $lnYzjg_t$ | 0.0877 | 0.0032 | 0.9019 |
| $lnSzjg_t$ | 0.0000 | 0.0000 | 0.0001 |
| $lnWzf_t$ | 0.0000 | 0.0000 | 0.2276 |
| $lnRgcb_t$ | 0.1598 | 0.0000 | 0.0109 |
| $lnTdcb_t$ | 0.0000 | 0.0001 | 0.9506 |

**Table 5. Cointegration test results.**

|  | Cointegration test | Statistic | p-value |
|---|---|---|---|
| KAO test | Modified Dickey-Fuller t | 1.7437 | 0.0406 |
|  | Dickey-Fuller t | 2.2654 | 0.0117 |
|  | Augmented Dickey-Fuller t | 3.0665 | 0.0011 |
|  | Unadjusted modified Dickey-Fuller t | 1.2720 | 0.1017 |
|  | Unadjusted Dickey-Fuller t | 1.7936 | 0.0364 |
| Pedroni test | Modified Phillips-Perron t | 6.2019 | 0.0000 |
|  | Phillips-Perron t | -13.4162 | 0.0000 |
|  | Augmented Dickey-Fuller t | -1.530e+15 | 0.0000 |
| WesterLund test | Variance ratio | 5.8547 | 0.0000 |

pig production, and analyze the previous period's price as an instrumental variable. This paper refers to this setting, setting $lnSzjg_t$ as an endogenous variable and $lnSzjg_{t-1}$ as an instrumental variable. In order to avoid the endogenous problem between environmental regulations and the slaughter of pigs, the method of introducing $lnSzjg_{t-1}$ is used for reference to construct instrumental variables of environmental regulations, variable $lnHjgz_t$ is lagging processing, and $lnHjgz_{t-1}$ is used as an instrumental variable for analysis. In order to test the hysteresis of the impact of environmental regulations on pig production, the variable $lnHjgz_{t-2}$ is specially introduced, Regression 3 is constructed, and test the nonlinear impact of environmental regulations in Regression 4 and 5. The regression results are shown in Table 6.

In Regression 3, the regression coefficient of variable $lnSzcl_{t-1}$ is 0.7136, and it is significant at the 1% level, indicating that the pig production in the t period is affected by the pig production in the t-1 period, that is, there is "inertia" in pig production. From the perspective of capital, Wang G et al. [30] believe that the "inertia" is due to the entry and exit barriers of thebehavior of the producers. The regression coefficients of variables $lnWzf_t$ and $lnRgcb_t$ are not significant below the 5% level. Variable $lnSzjg_{t-1}$ is significantly positive at the 10% level, and variable $lnTdcb_t$ is significantly negative at the 1% level, indicating that the slaughter of pigs is affected by land costs and price expectations. But it is not affected by material service fees and labor costs. This result verifies the view put forward by Wang H et al. [31] that good price expectations and stable cooperation are positively affecting the slaughter of pigs. The coefficient of the variable $ASF$ is -0.1106, and it is significant at the 1% level. The outbreak of African swine fever will reduce the number of pigs for slaughter by 11.06%. The coefficients of variables $lnHjgz_{t-1}$ and $lnHjgz_{t-2}$ are 0.0556 and -0.0431, respectively, which are both significant at the 5% level, which proves Hypothesis 1. In Regression 4, the regression coefficients of variable $lnHjgz_{t-1}^2$ and variable $lnHjgz_{t-1}$ are not significant at the 5% level. In Regression 5, the regression coefficients of variable $lnHjgz_{t-2}^2$ and variable $lnHjgz_{t-2}$ are 0.0642 and -0.9653, respectively, and are significant at the 1% level. Combined with the regression coefficients of variables $lnHjgz_{t-1}$ in Regression 3, it can be seen that there is a lag effect in the impact of environmental regulations on the slaughter of pigs. That is, in the short term, the strengthening of environmental regulations will increase the slaughter of pigs in short term, but in the long term it will show a positive U-shaped relationship. When the intensity of environmental regulations is lower than 2286.50, the hysteresis effect of environmental regulations is shown to inhibit the slaughter of pigs. When the intensity of environmental regulations is higher than 2286.50, the hysteresis effect of environmental regulations will also increase the slaughter of pigs. When the intensity of environmental regulation is low, environmental regulation will be strengthened through policies such as "prohibition" and "restriction" of produce. In the short term, the production capacity of pigs will be cleared, and the producers will produce a large

**Table 6. Empirical analysis results of the impact of environmental regulations on pig production.**

| | | Regression 1 (One-step method) | Regression 2 (Two-step method) | Regression 3 (Endogenous estimate) | Regression 4 (Endogenous estimate) | Regression 5 (Endogenous estimate) |
|---|---|---|---|---|---|---|
| $lnSzcl_{t-1}$ | coef. | 0.9827*** | 0.9811*** | 0.7136*** | 0.9054*** | 0.9873*** |
| | t | 76.04 | 66.73 | 4.10 | 12.01 | 42.20 |
| $lnHjgz_t$ | coef. | 0.0142*** | 0.0128 | 0.0186 | | |
| | t | 4.36 | 1.06 | 1.29 | | |
| $lnHjgz_{t-1}^2$ | | | | | 0.0149 | |
| | | | | | 1.02 | |
| $lnHjgz_{t-1}$ | coef. | | | 0.0556** | -0.1908 | |
| | t | | | 2.35 | -0.88 | |
| $lnHjgz_{t-2}^2$ | | | | | | 0.0642*** |
| | | | | | | 3.22 |
| $lnHjgz_{t-2}$ | coef. | | | -0.0431** | | -0.9653*** |
| | t | | | -2.03 | | -3.23 |
| $lnSzjg_t$ | coef. | -0.0582*** | -0.0567** | -0.1117*** | -0.0669*** | -0.0455*** |
| | t | -6.13 | -2.41 | -4.08 | -5.29 | -3.15 |
| $lnSzjg_{t-1}$ | coef. | | | 0.0729** | 0.0482*** | 0.0742*** |
| | t | | | 2.02 | 4.40 | 9.82 |
| $lnWzf_t$ | coef. | 0.0132*** | 0.0136 | 0.0071 | 0.0043 | -0.0014 |
| | t | 6.02 | 0.94 | 1.32 | 0.59 | -0.15 |
| $lnRgcb_t$ | coef. | 0.0036 | 0.0040 | 0.0446 | -0.0097 | -0.0206** |
| | t | 0.53 | 0.32 | 1.14 | -0.72 | -2.23 |
| $lnTdcb_t$ | coef. | -0.0262*** | -0.0264*** | -0.0182*** | -0.0226*** | -0.0333*** |
| | t | -13.30 | -3.86 | -4.06 | -3.96 | -5.54 |
| ASF | coef. | -0.1398*** | -0.1420*** | -0.1106*** | -0.1391*** | -0.1726*** |
| | t | -23.42 | -5.94 | -4.89 | -14.66 | -11.51 |
| Abond test | AR2 | -0.6991 | -0.7192 | -0.5790 | -0.2182 | 0.0926 |
| | P | 0.4845 | 0.4720 | 0.5626 | 0.8273 | 0.9262 |
| Sargan test | Chi2 | 27.56 | 182.25 | 21.69 | 23.47 | 25.87 |
| | p | 1.0000 | 0.0000 | 1.0000 | 1.0000 | 1.0000 |

Note: *, **, *** indicate significant at the 10%, 5% and 1% significance levels, respectively.

number of pigs for slaughter, and short-term supply will increase. However, with the reduction of pig production capacity, long-term pig production has been insufficient, showing a long-term inhibitory effect on production. When the environmental regulation is higher than the critical value, the environmental regulation can improve the production efficiency of the producers through the innovation compensation effect, showing the production incentive effect during the production period.

## 4.3. Analysis of mediating effect of herd structure

To test Hypothesis 2 and Hypothesis 3, the mediation effect model constructed above is used to test the mediation effect of the pig herd structure in the relationship between environmental regulation and pig production, and to control other factors that affect pig production. Due to the lag effect of environmental regulation on pig production, the variables $lnHjgz_t$ and $lnHjgz_{t-1}$ are used in the regression model to perform regression to test the mediating effect of environmental regulation at the stage difference. In order to avoid the mutual cause and effect between the herd structure and the $lnYzjg_t$ slaughter of pigs, $lnYzjg_t$ is processed afterwards, and the

**Table 7. Optimal herd structure and U-shaped moderating effect analysis results.**

| | | Regression 6 $lnYzjg_t$ | Regression 7 $lnYzjg_t$ | Regression 8 $lnYzjg_t$ | Regression 9 $lnSzcl_t$ | Regression 10 $lnSzcl_t$ |
|---|---|---|---|---|---|---|
| $lnSzcl_{t-1}$ | coef. | | | | 0.6616*** | 1.0081*** |
| | t | | | | 4.94 | 54.16 |
| $lnYzjg_{t-1}^2$ | coef. | | | | | -1.3436*** |
| | t | | | | | -2.70 |
| $lnYzjg_{t-1}$ | coef. | 0.4106** | 0.5310** | 0.6246*** | 0.5180*** | -5.9510*** |
| | t | 2.34 | 6.86 | 6.22 | 3.25 | -2.58 |
| $lnHjgz_t^2$ | coef. | | 0.0267*** | | | |
| | t | | 4.07 | | | |
| $lnHjgz_t$ | coef. | 0.0186*** | -0.3607*** | | -0.0107 | |
| | t | 2.89 | -3.72 | | -0.49 | |
| $lnHjgz_{t-1}^2$ | coef. | | | 0.0385*** | | |
| | t | | | 3.18 | | |
| $lnHjgz_{t-1}$ | coef. | 0.0371*** | | -0.5340*** | 0.0651** | 0.0383*** |
| | t | 3.03 | | -2.99 | 2.49 | 4.93 |
| $lnHjgz_{t-1}^2$ | coef. | | | | | 0.0725*** |
| | t | | | | | 3.84 |
| $lnHjgz_{t-2}$ | coef. | | | | -0.0878*** | -1.1247*** |
| | t | | | | -2.84 | -3.91 |
| $lnSzjg_t$ | coef. | -0.0222 | 0.0137* | 0.0225** | -0.1223*** | -0.0383* |
| | t | -1.20 | 1.65 | 2.44 | -5.12 | -1.83 |
| $lnSzjg_{t-1}$ | coef. | 0.0115** | 0.0254*** | 0.0110** | 0.1092*** | 0.1324*** |
| | t | 2.54 | 5.54 | 2.31 | 4.10 | 8.80 |
| $lnWzf_t$ | coef. | -0.0154*** | -0.0259*** | -0.0288*** | 0.0031 | -0.0145*** |
| | t | -2.69 | -8.12 | -6.55 | 0.34 | -2.74 |
| $lnRgcb_t$ | coef. | 0.0193** | 0.0068 | 0.0007 | 0.0325 | -0.0661*** |
| | t | 2.28 | 0.97 | 0.12 | 1.25 | -3.61 |
| $lnTdcb_t$ | coef. | -0.0124*** | -0.0089** | -0.0139*** | -0.0117*** | -0.0125*** |
| | t | -3.04 | -2.40 | -3.04 | -2.79 | -9.14 |
| $lnASF$ | coef. | 0.0797*** | 0.0295** | 0.0463*** | -0.1355*** | -0.1560*** |
| | t | 5.12 | 2.16 | 5.30 | -4.61 | -9.14 |
| Abond test | AR2 | 1.2773 | 1.3207 | 1.2218 | 0.2985 | -0.7622 |
| | P | 0.2015 | 0.1866 | 0.2218 | 0.7653 | 0.4459 |
| Sargan test | Chi2 | 24.86 | 18.93 | 18.68 | 13.63 | 23.3138 |
| | p | 1.0000 | 1.0000 | 1.0000 | 1.0000 | 0.9344 |

Note: *, **, *** indicate significant at the 10%, 5% and 1% significance levels, respectively.

system GMM is estimated with $lnYzjg_{t-1}$ as an instrument variable. The results are shown in Table 7.

In Regression 6, the coefficients of the variable $lnHjgz_t$ and the variable $lnHjgz_{t-1}$ are 0.0186 and 0.0371 respectively, and both are significant at the 1% level, indicating that there is a hysteresis effect in the impact of environmental regulations on the herd structure, and it is necessary to test whether there is a non-linear effect in stages, and construct Regression 7 and Regression 8. In Regression 7, the coefficients of the variable $lnHjgz_t^2$ and the variable $lnHjgz_t$ are 0.0267 and -0.3607, respectively, and both are significant at the 1% level. Comparing the coefficients of the variable $lnHjgz_{t-1}^2$ and the variable $lnHjgz_{t-1}$ in Regressions 8, it can be seen that the impact of environmental regulations on the herd structure is U-shaped. And with the

increase of time, the marginal impact of environmental regulations on the aquaculture structure will increase, but it still shows a U-shaped relationship, that is, with the increase of time, the marginal impact of environmental regulations on the aquaculture structure will gradually enlarge. In Regressions 6, 7, and 8, the coefficients of the variable *ASF* are always positive and significant at the 5% level, indicating that the farmers will give priority to reducing pigs when reducing production capacity after the outbreak. This result also provides factual support for the assumptions of constructing the model.

According to the regression coefficients of variable $lnYzjg_{t-1}$ and variable $lnYzjg_{t-1}^2$ in Regression 9 and Regression 10, we can see that there is a non-linear relationship between the herd structure and the slaughter of pigs. Therefore, further analysis is based on Regression 10. In Regression 10, the coefficients of the variable $lnYzjg_{t-1}^2$ and the variable $lnYzjg_{t-1}$ are both negative and significant at the 1% level, and the variable $lnYzjg_{t-1}$ has an inverted U-shaped relationship with the variable $lnSzcl_t$, and its symmetry axis is -2.2146. That is, when the herd structure level is 0.1092, the slaughter of pigs reaches a maximum value, and there is an optimal herd structure. Hypothesis 2 is proved. Only analyzing the optimal herd structure has limited guiding effect on policy regulation. It is necessary to conduct a comparative analysis based on the current status of pig herd structures in various regions. The results are shown in Fig 5.

Comparing the herd structure in the existing provinces in China, in 2018, only Liaoning, Inner Liaoning, Hainan and Xinjiang are located on the right side of the symmetry axis of the inverted U-shaped curve, and the proportion of pig production needs to be adjusted appropriately. The remaining provinces selected for the sample are all located below the optimal herd structure, and the proportion of sows in the stock should be appropriately increased to ensure pig production.

Combining Regression 7 and Regression 10, we can see that there is a U-shaped relationship between environmental regulations and herd structure, and there is an inverted U-shaped relationship between herd structure and pig production, that is, the herd structure has a mediating effect in the impact of environmental regulations on pig production. Hypothesis 3 is proved. According to the regression coefficient analysis, it is found that when the coefficient efficiency of the variable $lnHjgz_t$ is 11.21, the mediation effect of the herd structure will promote the slaughter of pigs. Combined with the current situation of environmental regulations, it can be seen that under the current intensity of environmental regulations, this mediating effect will only amplify the promotion of environmental regulations on the slaughter of pigs.

## 4.4. Further analysis: Early warning of production fluctuation based on herd structure

In order to further analyze whether the herd structure can warn the fluctuation of pig slaughter, this paper first conducts a feasibility analysis from two aspects of the forecasting ability and the index setting process, and compares the estimation methods of the pork-grain ratio, and gives an early warning interval.

In terms of predictive ability, pig stock and sow stock are used to predict the short-term and mid-to-long-term slaughter of pigs, respectively. L.M. Plà et al. [32] build a Markov model based on the herd structure to provide production decisions for pig companies. In addition, in Regression 9, the significant difference between the variable $lnYzjg_{t-1}$ and the variable $lnYzjg_t$ indicates that the herd structure will affect the number of pigs that are lagging in the first period. It can be considered that the herd structure has certain predictability in identifying fluctuations in the slaughter of pigs. Reproductive sows and pigs have decreased in the same proportion. For example, affected by African Swine Fever, China's pigs and reproductive sows decreased by 38.7% and 37.4% respectively in August 2019 [33]. The herd structure has no

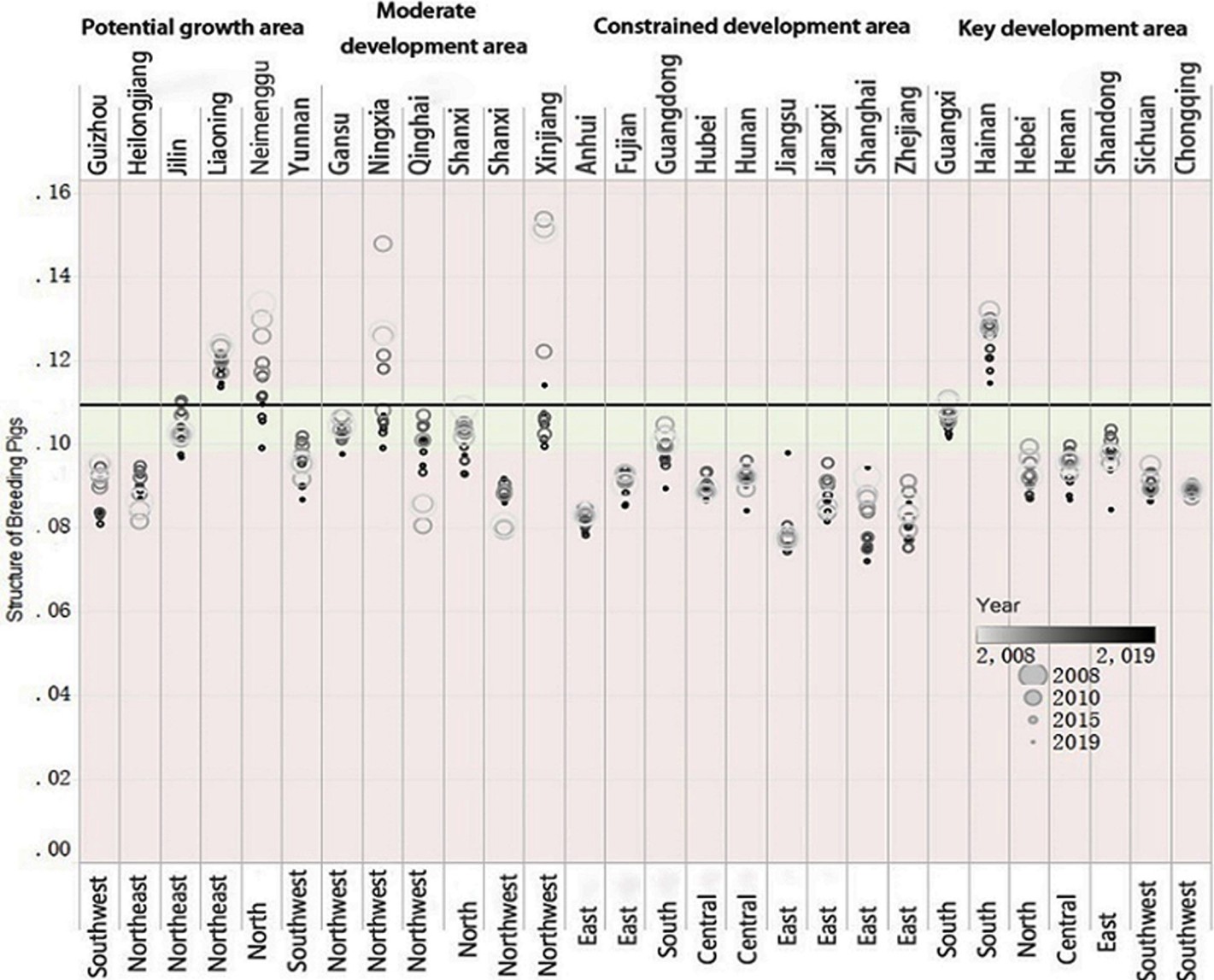

**Fig 5. Schematic diagram of adjustment of herd structure in various provinces.**

significant impact, but it still leads to a shortage of pigs in the market. It has no significant impact on the overall herd structure, but it still leads to a shortage of pigs in the market. The herd structure can be used as a supplementary monitoring indicator to stabilize the supply of pigs, combined with the rate of change in the stock of reproductive sows, to provide early warning of fluctuations in pig production. Among them, the rate of change in the stock of reproductive sows focuses on predicting whether the supply of pigs can meet market demand, and the herd structure focuses on whether the supply of pigs will produce large supply fluctuations. In the indicator setting process, this paper compares the process of setting the pork-grain ratio as an early warning indicator for pig price fluctuations.

Combining existing studies, it is found that as early as 1985, there was research and analysis of the impact of the pork-grain price ratio. By 2020, a relatively reliable forecasting system and

method have been formed. For example, Chen Juhong [34] uses the new multiple range test method to test the impact of the pork-grain price ratio and pork price fluctuations and sets an early warning interval of 5.0 to 5.5. Compared with the research on the pork-grain ratio, there are less researches on the herd structure, so this paper will only make a brief analysis here. According to the stock of reproductive sows and pigs monitored by the fixed observation points of the Ministry of Agriculture, the quarterly herd structure in China is estimated. The specific results are shown in Fig 6.

It can be seen from the figure that there are obvious seasonal fluctuations in the herd structure and the number of pigs. The HP filtering method is used to remove seasonal fluctuations, and the herd structure $Yzig^*$ and the slaughter of pigs $Szcl^*$ after removing the seasonal fluctuations are obtained. Taking into account the growth cycle of fattening pigs and reproductive sows, half a year and one year are selected as the short-term and long-term time ranges for the analysis of slaughter fluctuations, and the difference between short-term and long-term fluctuations in slaughter changes is constructed as $Dis^* = Szcl^*_{t+4} - Szcl^*_{t+2}$, where $^*$ means removing seasonal fluctuations, $Szcl_{t+4}$ is the amount of pigs to be slaughtered one year later, $Szcl_{t+2}$ is the output of pigs after six months. According to the threshold model structure, there is a breakpoint when the herd structure is 0.0980, and the variable $Yzjg^*$ in the overall regression result will positively affect the variable $Dis^*$ at the 5% significance level. According to the calculation of the coefficient, when the variable $Yzjg^*$ is 0.1058, the variable $Dis^*$ is 0, that is, there will be no long-term and short-term fluctuations. Early warning is carried out according to the threshold value, and the early warning interval is shown in Table 8.

When the herd structure is less than or equal to 0.0980, although the pig industry can guarantee short-term pig supply, there is a long-term supply shortage risk; when the herd structure is greater than 0.0884 but less than or equal to 0.1135, the pig industry can maintain a stable supply; when the herd structure is greater than or equal to 0.1135, the production capacity of the pig industry will increase in the long term, but it may face the risk of short-term supply shortage.

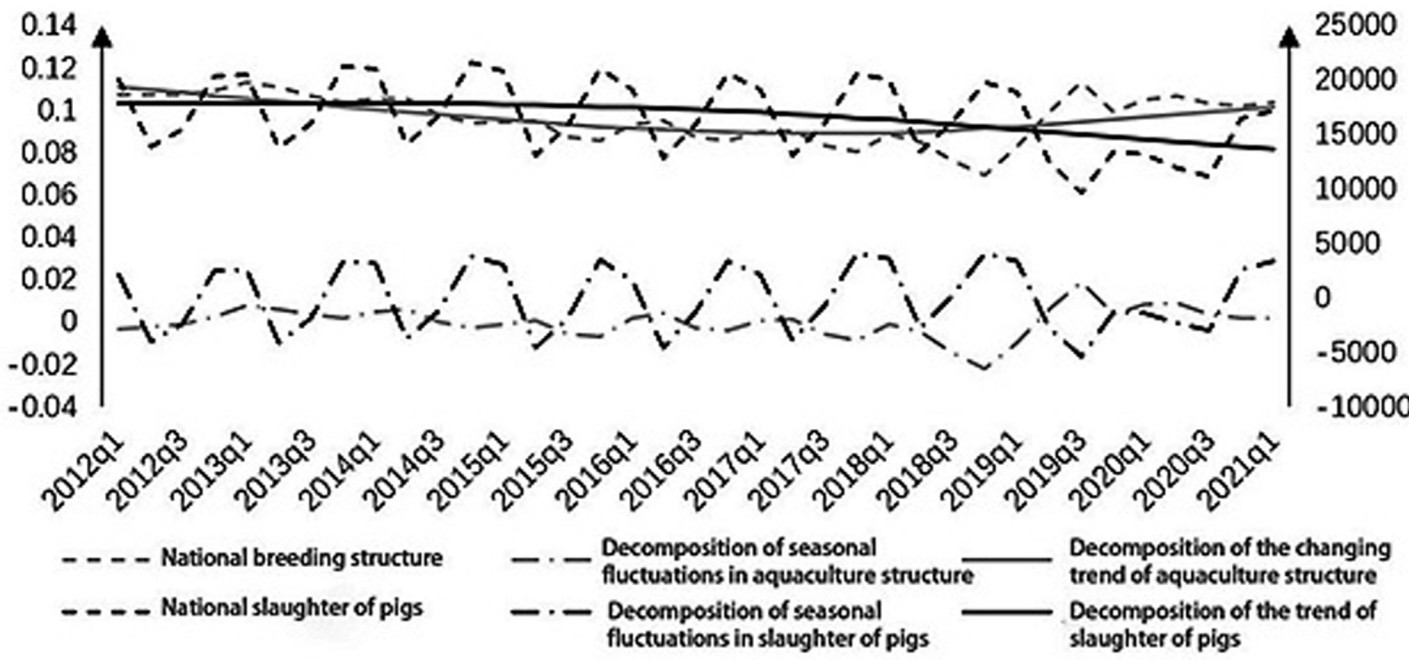

**Fig 6. Trend of China's pig price and herd structure change and HP breakdown.**

**Table 8. Early warning interval of herd structure.**

| Herd structure value | Interval type |
|---|---|
| $Yzjg \leq 0.0980$ | Long-term shortage warning |
| $0.0980 < Yzjg \leq 0.1135$ | Stable supply interval |
| $0.1135 \leq Yzjg$ | Short-term shortage warning |

## 5. Conclusions and discussion

### 5.1. Conclusions and contributions

Scholars based on the static point of view believe that environmental regulations will increase the production cost of aquaculture enterprises and reduce pollution by reducing production. However, as the main source of meat products in China, insufficient supply will cause a series of social problems. In Porter's hypothesis and innovation compensation theory, it is believed that environmental regulation can stimulate the innovation ability of enterprises, realize the compensation for the cost of pig production enterprises, and thus stabilize production. Under the two theories, there is no clear basis for the formulation of environmental regulation policies for pig farming. At this time, studying the impact of environmental regulation on pig production is helpful to formulate more accurate environmental regulation policies. We use the organic composition theory of capital for reference, take the community structure as the breakthrough point, analyze the impact of environmental regulation on pig production, and whether we can adjust the impact of environmental regulation on pig production through the community structure, and forecast the fluctuation of pig production in a certain period. The study found that the herd structure is affected by environmental regulations and has a significant impact on pig production. The herd structure can be used as a supplementary monitoring indicator to stabilize the supply of live pigs, and can be used to warn the fluctuation of pig production in combination with the change rate of sow. In the short term, strengthening environmental protection regulations can increase the number of live pigs on the market, but in the long term, the impact of environmental regulations on the number of live pigs on the market is U-shaped. The theoretical and empirical contributions of this paper are as follows:

First, our research complements the environmental regulation theory and applies it to the pig market in China. In the existing regulation theory, environmental regulation mainly includes following the cost theory and Porter hypothesis. This paper uses Marx's organic composition of capital for reference, and takes the herd structure as an intermediary, and incorporates it into the model of environmental regulation affecting pig production. The results of empirical analysis show that the herd structure is affected by environmental regulations and will also affect pig production. This discovery provides a new starting point for the formulation of environmental regulation policies, namely the industrial adjustment method based on the herd structure. The method can slow down the inhibition effect of environmental regulations on production by adjusting the proportion of sows and fattening pigs in the pig industry, and give consideration to environmental protection and pork supply.

Second, our research verified the regulation and early warning function in the production of live pigs with herd structure. At present, the existing pig early warning system is based on the pig-to-grain ratio. In the "Implementation Plan for Regulation and Control of Pig Production Capacity (Provisional)" dated September 23, 2021, the core regulation index is the change rate of sows capable. However, the supply early warning based on the change rate of sows capable can only predict the change of absolute supply. This system is not sensitive to fluctuations in supply, especially after multiple iterations, the early warning effect on fluctuations is often unrealistic. Based on the analysis of herd structure as an intermediate variable, this paper

shows that under a certain level of environmental regulation, there is an optimal herd structure to maximize the slaughter of live pigs, and further analysis results show that there is no significant fluctuation in the slaughter of live pigs corresponding to a specific range of herd structure. The early warning of herd structure is actually a balance between short-term supply and long-term supply of live pigs, which is more advantageous than the existing early warning system in the aspect of early warning supply fluctuation. Based on the above viewpoints, the development of pig industry can be adjusted by adjusting the macro herd structure. For example, the proportion of sows capable in the macro herd structure can be increased by increasing the purchase and elimination subsidies of sows capable by farmers, so as to realize the expanded production of pig production.

Third, our research combed the environmental regulation policies of China's pig industry from 2008 to 2019, and based on rational policy assessment, constructed the effectiveness indicators of environmental regulation policies only for pig industry from the four angles of objective, measure, feedback and intensity. This attempt not only effectively quantifies the intensity of environmental regulation policies, but also provides a reference for analyzing the policy effectiveness of specific industries.

## 5.2. Limitations and future research directions

This article has the following short comings: First, only the panel data of a single country, China, are used, and it is worthwhile to further explore the industrial situation of countries with high degree of pig industrialization, such as the United States, Germany, Denmark, etc. Second, due to the limitation of data availability, in this paper, we use virtual variables as the epidemic situation in pig production environment. Although we consider the impact of external shocks, this proxy variable still has limitations and needs further improvement. Third, this paper only focuses on the intermediary role of herd structure in environmental regulation affecting pig production, but this research has not been carried out on the existence and mechanism of other intermediaries. Future research can further explore the intermediary role between regulation of environment and pig production.

## Author Contributions

**Conceptualization:** Gangyi Wang, Yuzhuo Shen.

**Data curation:** Yuzhuo Shen, Chunlei Li.

**Formal analysis:** Gangyi Wang, Yuzhuo Shen.

**Funding acquisition:** Gangyi Wang.

**Investigation:** Yuzhuo Shen.

**Methodology:** Yuzhuo Shen.

**Software:** Yuzhuo Shen.

**Supervision:** Gangyi Wang.

**Writing – original draft:** Yuzhuo Shen.

**Writing – review & editing:** Gangyi Wang, Qiuping Zhu, Aidyn ZhanBota.

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
