## [Decision Letter · Decision Letter 0]

15 Nov 2021

PONE-D-21-31548The regulatory effect of breeding pig herd structure on pig production under the environmental regulationPLOS ONE

Dear Dr. Zhanbota,

Thank you for submitting your manuscript to PLOS ONE. After careful consideration, we feel that it has merit but does not fully meet PLOS ONE’s publication criteria as it currently stands. Therefore, we invite you to submit a revised version of the manuscript that addresses the points raised during the review process. Please submit your revised manuscript by Dec 30 2021 11:59PM. If you will need more time than this to complete your revisions, please reply to this message or contact the journal office at plosone@plos.org. Please include the following items when submitting your revised manuscript:A rebuttal letter that responds to each point raised by the academic editor and reviewer(s). You should upload this letter as a separate file labeled 'Response to Reviewers'.A marked-up copy of your manuscript that highlights changes made to the original version. You should upload this as a separate file labeled 'Revised Manuscript with Track Changes'.An unmarked version of your revised paper without tracked changes. You should upload this as a separate file labeled 'Manuscript'.

We look forward to receiving your revised manuscript.

Kind regards,

Adnan Noor Shah, PhD

Academic Editor

PLOS ONE

Journal Requirements:

"No"

5. Please ensure that you refer to Figure 3 in your text as, if accepted, production will need this reference to link the reader to the figure.

Reviewers' comments:

Reviewer's Responses to Questions

**Comments to the Author**

1. Is the manuscript technically sound, and do the data support the conclusions?

Reviewer #1: Yes

Reviewer #2: Yes

2. Has the statistical analysis been performed appropriately and rigorously? 

Reviewer #1: Yes

Reviewer #2: Yes

3. Have the authors made all data underlying the findings in their manuscript fully available?

Reviewer #1: Yes

Reviewer #2: Yes

4. Is the manuscript presented in an intelligible fashion and written in standard English?

Reviewer #1: No

Reviewer #2: Yes

5. Review Comments to the Author

Reviewer #1: To my opinion, your topic is principally interesting and important for the scientific community and eyond.

As I am not a native speaker, I cannot judge the linguistic English quality of the manuscript in every detail. However, I think there is some room for improvement, e.g. in line 142 ff. Another example: in line 170/171, I would write “…believe that they have…” instead of a repetition of “environmental regulations and village regulations and folk conventions”.

Since I am primarily concerned with other methods (life cycle assessment and sustainability analyses), I cannot give a well-informed opinion on the methodological novelty of the study. The calculations and the statistical seem correct.

Your manuscript does not follow the classical structure proposed by scientific journals such as (alos) PLOS One (with Introduction, Material & Methods, Results, Discussion,…). I do not know if this is obligat

Regarding the abstract:

Line 8 (and other places!): Why do you address pollution control only regarding breeding? I know that your modeling focusses on breeding, but at least in European and North American pig production, the fattening phase is the most relevant point, accounting for about three-fourths of the impact over the life cycle (in terms of GHG emissions, among other impacts). Thus, I would rather write “pig production” (or “pig husbandry”?) instead of “breeding” in the Introduction schapter.

Line 19: It is difficult to interpret the two early warning values without the description of any unit.

Regarding Introduction:

The chapter well describes the Chinese situation (the application case), however, in comparison to other papers from my discipline, I would have expected some more references to peer reviewed literature and a description of, for instance, another country/continent or the global situation concerning price fluctuations, pests such as the African Swine Fever, etc. To my knowledge, the Chinese pork market is related to the markets of many other countries with specific mass and price fluctuations. Other countries have their own pollution control regulations – how are they interconnected and comparable with Chinese regulations? These aspects are not reflected in your manuscript! But they could be at least considered in a part of your Introduction section and taken up in the Discussion section.

Regarding Material and Method section:

The first part(s) of your rather long Material and Method section (“Literature review”) seems to me as a part of the Introduction. You wrote a research article, not a review paper.

In my opinion the Material and Methods section in a research article needs to describe all important aspects, which form the basis for your own modelling. To my opinion, the (narrower) Material and Method section should describe the used databases and key words regarding “Literature research” as well as data sources, models (formulas,...) and the variables that are used in the analysis. Your Material and Method section (“Literature review”) describes over long passages more the background of your work.

The chapter (sub-/heading) structure of the manuscript, especially regarding Material & Methods, and the size of some chapters seems somewhat disproportionate.

Line 106: You use the abbreviation “CPI” for one time only. Please write it in full words.

Line 123: The same with the abbreviation “DID”. Please write it in full words.

Furthermore, some abbreviations regarding statistical analysis (i.a. LCC, HP, IPS test) are not given with a full name by the first use.

I like the message from lines 176 to 189 and I think that this would also fit well at the end of the Introduction chapter. Then, before the actual chapter Material & Methods, formulate the hypotheses or research questions!

I miss a description of the early warning (score) in Material & Methods. This is a value that you have specifically calculated within your paper, isn’t it? Thus I would like a description of how you have used it.

The description of the Material & Methods chapter (“Theoretical Analysis”; from line 190 onwards) could probably be phrased more concisely (and more "scientifically"), especially from line 205 onwards. For instance, I would delete the sentence with “Although there is a saying that…” or radically shorten it!

Line 257-258: Please rephrase the title, e.g. “Pig production factors and environmental regulation” (Is that too much shortened? In any case, I wouldn't use a question-like sentence as a heading).

Generally, I miss a numbering of the headings in your manuscript in order to be able to clearly distinguish their levels.

Line 273, 306 and others: you did not define hypotheses (nor detailed research questions) in the Introduction section, but describe the hypotheses' achievement in the Material & Methods-section (what is not possible!).

For me, the first sentence in the Conclusions is one of those sentences that needs to be simplified. Such long and convoluted sentences occur sometimes throughout the document, but make your manuscript difficult to read.

Reviewer #2: Comments to the Author: 1. From the perspective of the pig industry, this article identifies the dynamic impact of environmental regulations on pig production in China, which has an academic value.

2. However, this article has some technical datils to improve. ①The variable which is utilized to measure breeding herd structure is a little simple. I do not think it can proxy the structure precisely. ②The control variables in sys-GMM are the price which contains 12 years, but this article did not adjust them with price index. It may lead to estimation bias. ③The variable lnASF is not shown in the variables descriptive statistics table. ④what is the meaning of three stars on t statistics value of variable lnASF in Regression 3 in Table 6？

6. PLOS authors have the option to publish the peer review history of their article (what does this mean?). If published, this will include your full peer review and any attached files.

Reviewer #1: No

Reviewer #2: No

---

## [Author Response · Author response to Decision Letter 0]

9 Mar 2022

Dear reviewers

We have revised it according to your comments. In order to facilitate your cross-reference, we have uploaded three manuscripts, namely The First Submitted Manuscript, The Revised Manuscript, and The Revised Manuscript with Traces Attached. 

Finally, we would like to thank you again for your work.

Yuzhuo Shen

February 7, 2022

---

## [Decision Letter · Decision Letter 1]

25 Mar 2022

The regulatory effect of herd structure on pig production under the environmental regulation

PONE-D-21-31548R1

Dear Dr. Zhanbota,

We’re pleased to inform you that your manuscript has been judged scientifically suitable for publication and will be formally accepted for publication once it meets all outstanding technical requirements.

Kind regards,

Adnan Noor Shah, PhD

Academic Editor

PLOS ONE

Additional Editor Comments (optional):

Reviewers' comments:

Reviewer's Responses to Questions

**Comments to the Author**

1. If the authors have adequately addressed your comments raised in a previous round of review and you feel that this manuscript is now acceptable for publication, you may indicate that here to bypass the “Comments to the Author” section, enter your conflict of interest statement in the “Confidential to Editor” section, and submit your "Accept" recommendation.

Reviewer #1: All comments have been addressed

2. Is the manuscript technically sound, and do the data support the conclusions?

Reviewer #1: Yes

3. Has the statistical analysis been performed appropriately and rigorously? 

Reviewer #1: Yes

4. Have the authors made all data underlying the findings in their manuscript fully available?

Reviewer #1: Yes

5. Is the manuscript presented in an intelligible fashion and written in standard English?

Reviewer #1: Yes

6. Review Comments to the Author

Reviewer #1: (No Response)

7. PLOS authors have the option to publish the peer review history of their article (what does this mean?). If published, this will include your full peer review and any attached files.

Reviewer #1: No

---

## [Editor Report · Acceptance letter]

6 Apr 2022

PONE-D-21-31548R1 

The regulatory effect of herd structure on pig production under the environmental regulation 

Dear Dr. Zhanbota:

I'm pleased to inform you that your manuscript has been deemed suitable for publication in PLOS ONE. Congratulations! Your manuscript is now with our production department. 

Kind regards, 

on behalf of

Dr. Adnan Noor Shah 

Academic Editor

PLOS ONE